# Conotoxin Patenting Trends in Academia and Industry

**DOI:** 10.3390/md20080531

**Published:** 2022-08-19

**Authors:** Noemi Sanchez-Campos, Johanna Bernaldez-Sarabia, Alexei F. Licea-Navarro

**Affiliations:** 1Biomedical Innovation Department, Scientific Research Center and Higher Education from Ensenada (CICESE), Carretera Ensenada-Tijuana 3918, Zona Playitas, Ensenada 22860, BC, Mexico; 2Innovation and Development Office, Scientific Research Center and Higher Education from Ensenada (CICESE), Carretera Ensenada-Tijuana 3918, Zona Playitas, Ensenada 22860, BC, Mexico

**Keywords:** *Conus* patents, conotoxin patent, conopeptide patent, *Conus*, patent, Orbit, ziconotide

## Abstract

Sea snails of the genus *Conus* produce toxins that have been the subjects of numerous studies, projects, publications, and patents over the years. Since *Conus* toxins were discovered in the 1960s, their biological activity has been thought to have high pharmaceutical potential that could be explored beyond the limits of academic laboratories. We reviewed 224 patent documents related to conotoxins and conopeptides globally to determine the course that innovation and development has taken over the years, their primary applications, the technological trends over the last six years, and the leaders in the field, since the only previous patent review was performed in 2015 and focused in USA valid patents. In addition, we explored which countries/territories protect their inventions and patents and the most relevant collaborations among assignees. We also evaluated whether academia or pharmaceutical companies are the future of conotoxin research. We concluded that the 224 conotoxin patents reviewed in this study have more academic value than industrial value, which was noted by the number of active patents that have not yet been licensed and the contributions to medical research, especially as tools to study neuropathic pain, inflammation, immunology, drug design, receptor binding sites, cancer, neurotransmission, epilepsy, peptide biosynthesis, and depression. The aim of this review is to provide an overview of the current state of conotoxin patents, their main applications, and success based on the number of licensing and products in the market.

## 1. Introduction

Only a few established scientific groups around the world have dedicated themselves entirely to researching marine snails of the *Conus* genus. The largest group and pioneer in this field is led by Baldomero Olivera. In 1978, Olivera and colleagues described the complexity of *Conus* conotoxins for the first time when they discovered a peptide toxin from *Conus geographus*, a fish-hunting cone snail collected in the Philippines [1,2,3]. For over 40 years now, this group has been patenting and publishing information. The first conotoxin patent was submitted in 1982 by the Salk Institute for Biological Studies [4]. In this patent, Baldomero Olivera, Lourdes Cruz, William Gray, and Jean Rivier first disclosed the biological activity, chemical structures, and synthesis of three homologous toxic peptides. That initial discovery opened new avenues for research, which have been constructed molecule by molecule. One of the first avenues produced the first analgesic drug from a marine organism, ziconotide (ω-conopeptide MVIIA or SNX-111), which was initially isolated from *Conus magus* in 1982 [5].

In 1985, Olivera sent a calcium channel blocker, conopeptide, to George Miljanich, who wanted to test ω-conotoxins for his work on neurotransmitter release at the University of Southern California and, three years later, they published a book chapter together [6]. Miljanich continued the ω-conotoxins research [7,8] and the progression from pharmacological tools for basic neuroscience to a therapeutic candidate can be credited to him [9] since he was recruited by a private biotech start-up company called Neurex (Menlo Park, CA, USA) in 1988 and even when ω-conotoxins were initially considered useful as neuroprotective in animal models of stroke [10], Miljanich showed that they were potent, specific, and degradation resistant, making them perfect to become pharmaceutical compounds for pathological pain [11,12,13,14,15,16].

Meanwhile, Olivera and colleagues at the University of Utah needed to publish results to renew their grants, so they chose to patent some of the earliest conotoxin discoveries but, in doing so, lost ownership of the initial ω-conotoxins, as they had become part of the public domain. After this and to avoid using the university budget for patents, Cognetix Inc. (Salt Lake City, UT, USA, 1993) was founded by Olivera, and by 1997, the company had identified approximately 100 conotoxins, which were covered by granted patents or patent applications [12].

Cognetix Inc. synthesized over 200 derivatives of MVIIA, a ω-conotoxin that binds to a specific spinal cord region and that functions as an analgesic. Nonetheless, the original and natural product structure of ziconotide was chosen for clinical trials in 1993 by Neurex. However, given its chemical nature, it had to be delivered via intrathecal injection, which comes with various pros and cons [3,9,12].

In 1998, Elan Pharmaceuticals (Dublin, Ireland) acquired Neurex and led ziconotide through the remaining clinical trials. In December 2004, Elan Pharmaceuticals finally obtained FDA approval in the United States for ziconotide as a pain reliever and subsequently marketed it under the name of Prialt^®^ [3,12,17]. Ziconotide is a good example of the patent strategies that are used to prolong the protection of a product. The success of Prialt^®^ has led to different patents being filed to try to improve its administration route and its effect when combined with other analgesics [16].

Patents can provide a substantial amount of relevant information about specific technologies in the form of research trends, novelty, strengths, weaknesses, state of the art, collaborations, key players, successful and failed licensing, covered countries/territories, and markets just to mention a few. The only existing conotoxin patent review was performed in 2015 by two Australian researchers from the Institute for Molecular Bioscience of the University of Queensland in Brisbane [18]. These authors focused their search on the number of valid US conotoxin patents and reported that there were 38 valid US-issued patents as of June 2015. However, the authors mentioned that they only used the USPTO and WIPO patent databases, so they could not guarantee either completeness or accuracy of their data due to the complexity of the subject, as all patent information should be meticulously analyzed to obtain reliable information, it is necessary to use innovative patent mining technology tools to generate reliable and complete patent information for stakeholders, such as Orbit Intelligence by Questel (Paris, France) [19].

## 2. Results

Given that the objective of this study was to directly detect and quantify inventions related to conotoxins, we only focused on analyzing 224 patent documents. The remaining 137 patents were left out of the investigation. Although these 137 patents included conotoxins, they did not include any specific innovation related to novel conotoxin use, since they were used as research tools to test another inventions, which were not related to new and relevant conotoxin uses per se. All resulting images were obtained from Orbit Intelligence by Questel, and are reproduced under its permission.

### 2.1. Assignees, Inventors, and the Evolution of Patent Portfolio Sizes

After analyzing the selected patent documents, it became evident that conotoxin inventions have changed over the years. Although this technology was discovered in the 1960s, the first registered conotoxin patent was applied for in 1982 by the Salk Institute for Biological Studies [4]. This patent discloses the biological activity and chemical structures of three homologous toxic peptides and their synthesis for the first time. A few years later, this patent expired due to non-payment (lapsed) in 1988. In 1984, Ajinomoto applied for two patents for conotoxins to be used as a muscle relaxant [20] and anesthetic [21]. However, both applications were abandoned 10 and 11 years later, respectively, so they ceased to have effect.

Peaks in the number of patent applications during the last 41 years are apparent (Figure 1), with 15 patent applications in 2014; 12 in 2012 and 2019; 11 in 2000 and 2003; 10 in 2006, 2007, and 2017; and 9 in 2004, 2010, and 2017. Additionally, no patent applications occurred in 1983 and in the period between 1985 and 1988. Following this period, the lowest number of patent applications was registered from 1989 to 1992 and from 1994 to 1995 and in 1997, 1999, and 2021. When the number of applications decreases, a disengagement exists among field actors; however, a stable profile in terms of patent applications is a sign of sector maturity. It is also possible to distinguish peaks or troughs in the number of applications, which depend on patenting and R&D budgets and broader economic and strategic changes.

Figure 2 shows the evolution of conotoxin patent applications over time and shows the portfolio sizes of the applicants in the patent pool. From right to left, we can observe that the Salk Institute for Biological Studies is a pioneer applicant in this field with a 28-year-old patent portfolio of 5 applications, followed by a 5-year gap. After which, the University of Utah appears with a 23-year-old patent portfolio of 12 applications. Two 20-year-old patent portfolios can be observed next belonging to the University of Utah Research Foundation and Cognetix (29 and 16 patents, respectively). Then, a 19-year-old portfolio belonging to Xenome with 6 applications is apparent, followed by a 4-year gap. The 15-year-old portfolios of the University of Queensland and Zhejiang University can then be seen with 8 and 6 applications, respectively.

Both Sun Yat Sen University (18 applications) and the Institute of Biotechnology from the Academy of Military Medical Sciences of China (7 applications) have 12-year-old portfolios. Hainan University has a 10-year-old patent portfolio comprised of 13 applications. A 4-year gap can then be observed, which is followed by the 6-year-old portfolios of BGI/Beijing Genomics Institute (13 applications) and Tongji University (4 applications). The newest, 3-year-old patent portfolios belong to Kineta Chronic Pain and China Ocean University/Ocean University of China (4 applications each), followed by the Institute of Chemical Defense Chinese Academy of Military Sciences with its 2-year-old patent portfolio (3 applications).

Figure 2 is a good indicator of the level of inventiveness of all the active players, as it shows the top applicants (i.e., University of Utah Research Foundation, Cognetix, and Sun Yat Sen University) in terms of the number of conotoxin patents. This figure also shows new entrants or applicants and helps explain filing peaks (i.e., a notable number of applications are filed over a brief period), which could affect the global evolution of the filings. In this context, BGI represents a new, strong entrant from the last six years given its 13 patent applications. As the number of conotoxin patent applications has decreased over the last six years, Ocean University of China, Kineta, and the Institute of Chemical Defense, constitute a new, strong applicant group with 2- and 3-year-old portfolios that could be active for up to 17 additional years. A low average age for a small portfolio is typical of new entrants such as those of Ocean University of China, Kineta, the Institute of Chemical Defense, and Tongji University. However, a large portfolio with a low average age is indicative of new entrants (e.g., BGI) who invest heavily to position themselves in the analysis sector (Figure 2).

The top 20 assignees from 1981 to 2022 can be observed in Figure 3. Among these, the University of Utah Research Foundation stands out with almost consecutive patent applications between 1993 and 2017, the year of their last patent application. This entity also registered a high number of applications during one year (4 in 1996), along with Cognetix (4 in 2001), Sun Yat Sen University (4 in 2008), and Hainan University (5 in 2012). The record for the highest number of applications is held by BGI (11 patent applications in 2014; Figure 3).

Between 2016 and 2022 (Figure 4), Ocean University of China had the most patent applications (4), followed by Kineta (3), the Shenzhen Research Institute of Zhongshan (3), and the Institute of Chemical Defense Chinese Academy of Military Sciences (3). The University of Queensland, Tongji University, and Hainan University followed close behind with 2 applications each during the last 6 years. Finally, the University of Utah Research Foundation had only one application during this period (Figure 4).

One of the main objectives of this study was to detect active assignees with granted patents and to study the evolution of patent conotoxins during the last six years. Thus, we analyzed the top 20 applicants (Figure 5) and found that despite the high number of patent applications from the University of Utah Research Foundation, Sun Yat Sen University, Cognetix, and the University of Queensland, the majority of their patents had lapsed or were inactive due to their validity periods. We also found that Zhejiang University, the Salk Institute for Biological Studies, Jazz Pharmaceuticals International, Ajinomoto, and Azur Pharma International have no live patents. The University of Utah, Xenome, and Anygen appear ready to join this group because they only have one remaining live patent each. On the other hand, BGI stands out because all of its patents are granted and live. Other assignees, such as Hainan University and the Institute of Biotechnology from the Academy of Military Medical Sciences of China, have the majority of their patent applications granted with only a few that are now dead. New risers such as Kineta and Tongji University already have granted patents and pending applications, while Ocean University of China and the Institute of Chemical Defense Chinese Academy of Military Science have all of their applications still pending. The most recent pending applications belong to Kineta and the Institute of Chemical Defense of the Chinese Academy of Military Sciences (Figure 5).

### 2.2. Conotoxin Uses and Applications

The technology domains and the most important uses of the conotoxin patents from 1981 to June 2022 based on their International Patent Classification (IPC) code at submission are shown in Figure 6. We determined that the main uses of conotoxins are as pharmaceuticals (186 patents) and in biotechnology (59). Other uses included analysis of biological materials (36), basic materials in chemistry (12), and medical technology (8) as well as minor uses in the form of components in organic fine chemistry, food chemistry, surface and coating technology, chemical engineering, and computer technology (Figure 6).

### 2.3. Technology Trends of the Last 6 Years

The tendency of the primary uses in technology from 2016 to 2022 are shown in Figure 7. From 46 patents, we observed that the tendencies in technology have remained almost the same with Pharmaceuticals and Biotech on top (33 and 19 patents, respectively), followed by the analysis of biological materials (only 2) and basic materials chemistry (3). The patents claims reflected only one use of conotoxins in organic fine chemistry and one use in coating and surface technology, and their use in medical technology has now disappeared completely when compared to the patents filed in the past (Figure 7).

The technology domains of each one of the top 20 assignees are shown in Figure 8. The University of Utah Research Foundation patents cover the majority of the main known uses of conotoxins, with 25 pharmaceutical patents, 9 patents for analysis of biological materials, and 5 patents for biotechnology purposes. Other assignees differ slightly, and their patents cover the principal uses those of other applications such as basic material chemistry (Hainan University, four patents). In the case of Cognetix, these also include the protection of conotoxins for use in computer technology [22,23] (Figure 8).

### 2.4. Legal Status of Patent Applications

From 1981 to 2022, we found 90 active (alive) conotoxin patents and 134 that were already dead (inactive). Of the 90 live patents, 60 had been granted and 30 were still pending. Of the 134 dead or inactive patents, 23 had been revoked (invalidated patent and cancelled rights), 34 had expired (patents are valid for up to 20 years since their first filing application date), and 77 had lapsed (anticipated expiration due to non payment).

We also analyzed the legal status of the patent applications from the last 6 years (2016–2022, Figure 9) and found that there were 46 patent applications in total, 87% of which were live while 13% were dead. The live patent portion comprised 40 applications, including 15 that had been granted and 25 that were still pending (33% and 54%, respectively). The dead patent percentage contained only six applications, of which four had been revoked and two had lapsed (9% and 4%, respectively; Figure 9).

The first revoked patents correspond to applications submitted in 1993 and belong to the Brown University Research Foundation, the Salk Institute for Biological Studies, and Zeneca (Figure 10). The assignees of the top 20 group with revoked patents included Cognetix (1), Sun Yat Sen University (3), the Institute of Biotechnology from the Academy of Military Medical Sciences of China (1), Hainan University (1), the University of Queensland (1), the University of Utah Research Foundation (1), and Xenome (1). Of note, Shenzhen Research Institute of Zhongshan University had three patents revoked in a row that were submitted in 2016 [24,25,26]. In 2018, the last patent was revoked and belonged to Reali Tide Biological Technology Weihai [27] (Figure 10).

The analysis of expired patents by application year (Figure 11) indicated that 34 granted patents had lost their validity, some of which belonged to more than one assignee (joint ownership). The University of Utah Research Foundation, University of Utah, Cognetix, and Jazz Pharmaceuticals International stand out with 12, 8, 6, and 3 expired patents, respectively (Figure 11).

We observed a total of 77 lapsed patent applications (Figure 12). The assignees with the highest numbers of lapsed patents include Sun Yat Sen University (12), the University of Utah Research Foundation (10), and Cognetix (8; Figure 12).

### 2.5. Abandoned Patents

Along the way, many conotoxin patents were abandoned. We noticed that most patents expired due to non-payment (lapsed), regardless of whether they came from a private company or an academic institution. This may denote a possible lack of interest or a research and development (R+D)/Intellectual Property budget reduction.

### 2.6. Companies Interested in Conotoxins

Beside Neurex, other companies have also tried to exploit conotoxins. For example, Neuromed (Vancouver, BC, Canada) collaborated with Merck on a calcium channel blocker similar to Prialt^®^ but that was designed to be taken orally [28]. In Australia, Xenome Ltd. (Brisbane, Australia) advanced a new class of conopeptide for treating postoperative pain and chronic cancer pain until Phase II (discontinued), and Metabolic Pharmaceuticals tested a conopeptide for neuropathic pain that was discontinued because of lack of efficacy [9,29,30].

After Elan Pharmaceuticals (Ireland) bought Neurex for USD 760 million and led ziconotide through the remaining clinical trials in 1998 [9], Elan and Pfizer (formerly Warner-Lambert) developed ziconotide for the treatment of ischemia associated with head trauma and stroke in 1997 [10,14]. By April 1999, Parke-Davis (now Pfizer) was also working on the development of non-peptide analogs of ziconotide with the intention of producing orally available agents to treat chronic pain [17,31].

In January 2000 in Australia, Professor Paul Alewood, Professor Peter Andrews, Dr. Roger Drinkwater, and Dr. Richard Lewis, founded Xenome Ltd., a spin-off company from the University of Queensland that was launched to develop new pharmaceutical compounds based on venom peptides, which unfortunately did not succeed as expected. Even so, Xenome still has one granted patent, which will remain active until 2023 [32]; the remaining five have already expired [33,34,35,36,37].

To cover the European market, Eisai Co., Ltd. (Tokyo, Japan) obtained the exclusive rights from Elan Corp. in 2006 to develop, manufacture, and market Prialt^®^ as a non-opioid intrathecal infusion to treat severe chronic pain. In 2010, Azur Pharma Ltd. acquired Prialt^®^ for USD 12 million, and in 2012, the company merged with Jazz Pharmaceuticals, Inc. By 2014 and 2015, Prialt^®^ net sales reached USD 26.4 million. In 2018, Eisai transferred the exclusive development and marketing rights for Prialt^®^ in Europe to Riemser Pharma GmbH (Greifswald, Germany), which has since become the European distributor of the drug [38,39].

Of the companies that currently invest in conotoxin R+D, we only identified one (Kineta Inc., Seattle, WA, USA) that actively invests in conopeptides for pharmaceutical uses, specifically the treatment of chronic neuropathic pain. This company has its own R+D team that is supported by a strategic partnership with Genentech that was established in 2018. Currently, Kineta is testing one molecule in Phase I clinical studies to evaluate its safety, tolerability, and pharmacokinetics as a once-weekly, subcutaneous injectable therapy. The molecule is α-conotoxin KCP506, a long-acting α9α10 nicotinic acetylcholine receptor (nAChR) antagonist that has been developed to treat chronic neuropathic pain [40]. Kineta already has granted one patent family in Australia, Hong Kong, Israel, Japan, Taiwan, and the US, and it is still pending in Canada, China, the European Union, India, and Korea [41] additionally, Kineta has three pending patent applications [42,43,44].

An updated overview regarding conotoxins as drug leads and their current status in clinical trials or development stage is well documented in [29] and up to 2015 in [13].

### 2.7. Patent Licensing and Successful Products

Since the successful discovery of ziconotide and its subsequent approval by the FDA and the European Medicines Agency (EMA), no notable development adopted by the pharmaceutical industry has resulted in a conotoxin-related product. Of the products based on conopeptides, only two successful examples exist that have reached the market: XEP™-018 (anti-wrinkle use, Activen/Atheris) and Prialt^®^ (i.e., ziconotide, chronic pain treatment). However, as [9,29] mentioned, many conotoxins and conopeptides can be used as unique sources of diagnostic and research tools to study signaling pathways given their target specificity and in the particular case of ω-conotoxins, they have been used in more than 3000 published studies.

With regard to new licensing, a conotoxin insulin analog named “mini-Ins” is being developed from a full-potency insulin obtained from the fish-hunting cone snail *Conus geographus* and has been licensed to Monolog LLC (San Francisco, CA, USA) [45,46].

### 2.8. Principal Countries/Territories That Protect Patents

The top 20 countries/territories in which the assignees have submitted conotoxin patent applications that are still valid are shown in Figure 13. These countries/territories include (number of patent applications): China (63), USA (31), Germany (18), France (14), Switzerland (13), Canada (11), United Kingdom (11), Japan (11), Australia (10), India (8), Hong Kong (7), Korea (7), Brazil (6), Ireland (6), Israel (5), Mexico (4), Belgium (3), Singapore (3), Taiwan (3), and Iceland (2; Figure 13). Between 2016 and 2022, the countries/territories that submitted the most patent applications for conotoxin inventions were: China (29), the United States (9), Canada (6), Australia (6), and Japan (6), Brazil (5), India (4), Korea (4), Israel (3), Singapore (3), Hong Kong (2), Mexico (2), Russia (2), United Arab Emirates (1), Switzerland (1), Chile (1), Colombia (1), and Costa Rica (1; Figure 14).

### 2.9. The Most Relevant Collaborations

To analyze assignee collaborations, Orbit was set up to look for a minimum of one joint patent as a byproduct of collaboration (Figure 15 and Figure 16). After the initial patent of the Salk Institute in 1982 [4], Olivera and colleagues continued their conotoxin research. Between 1991 and 2017, they submitted 29 patents that included some findings from the University of Utah, the current affiliation of Olivera. Three of those 29 applications were also submitted in conjunction with the Salk Institute [47,48,49].

A patent application in 1998 included Cognetix, the Salk Institute, the University of Utah Research Foundation, and the Case Western Reserve University (Private, Cleveland, Ohio, USA) as assignees and was titled: “Interaction of alpha-conotoxin peptides with neuronal nicotinic acetylcholine receptors”. However, after its publication in the European Union, Japan, Australia, and Canada the patent was withdrawn and abandoned for unknown reasons [23].

David Craik and Paul Alewood, who were working with conotoxins in the University of Queensland in Brisbane, Australia, started patenting in 1998 [3]. Collectively, these two groups have three granted patents, one application that was left pending, and four others that are dead [33,37,50,51,52,53,54,55,56].

The most relevant patent holder assignees are shown in Figure 15. The collaboration between the University of Utah Research Foundation and the University of Queensland and the Max Planck Institute [53] is notable, as are diverse collaborations between the University of Queensland and the University of Sydney [56] and the University of Utah and the University of Missouri [57].

Figure 16 shows the collaboration between Xenome and the University of Queensland [33,37], Hainan Medical College and BGI [58,59], Tongji University and the University of Wollongong [60,61], Tongji University and RMIT University, Shanghai IBS, the Chinese Academy of Sciences [62], Shenzhen Huada Gene Institute, Shenzhen BGI Aquatic product technology [63], and Receptogen and Rumph Harold H [64], and the collaboration between Activen, CNRS, Atheris Laboratories, and Squirrel Laboratories Athens [65].

### 2.10. Current Leaders Based on Granted and Valid Patents

We identified current leaders in the field based on the number of granted and valid patents (Figure 5). We found that Sun Yat Sen University has 18 patents, but only 3 of these are granted and still valid; the remaining 15 are dead. At present, BGI from China is the assignee with the highest number of valid patents (13), all of which are granted and active. BGI is followed by Hainan University, which has eight granted patents, two pending applications, and three dead patents in its patent portfolio (Figure 5).

Assignees with the highest growth rates over the last six years based on the number of filings per year are Ocean University of China (400%), the Institute of Chemical Defense PLA (Chinese People’s Liberation Army), Academy of Military Science, Kineta Chronic Pain, Shenzhen Research Institute of Zhongshan University (300%), and Fujian Agriculture & Forestry University (200%; Figure 17). When the legal status of each assignee was also analyzed (Figure 18), Shenzhen Research Institute of Zhongshan University cannot be considered a top assignee, as all of its three patent applications were revoked right after publication. Therefore, Hainan Medical College can be considered the fifth top assignee from 2016 to 2021. The application years (between 2016 and 2022) for these top 20 assignees are shown in Figure 19.

## 3. Discussion

The technological landscape is nourished by patents through which technological trends can be analyzed and traced, but not all patents are issued or kept alive for sufficient time to be used or licensed. Patents also do not hold the same value to direct users and society in general; this depends on the approach and application of the patent. However, with time, some patents become more valuable to society than to the original assignee and may be used in different ways than expected [66].

Multiple means can be used to measure the success of scientific research projects that lead to product innovation. One of these methods is to establish the number of existing patents on the subject to determine which patents have been licensed, how many patents have been abandoned, how many patents have been granted but not exploited, and how many patents have been revoked. Conotoxin patents have been very active in terms of protecting Industrial Property, as many patent applications have been granted. However, when reviewing the number of successful products on the market (the true meaning of innovation is reaching and impacting the market), the field of conotoxins does not seem to be very successful at all. Only Prialt^®^ has achieved this kind of success, and this drug has passed through several companies before it was well positioned within the Pharmaceutical market.

Research and innovation are increasingly expected to provide solutions to societal challenges. Skeptic beliefs about the benefits and social accountability of science and technology are present, and society is often reproachful of these industries for not delivering on promises. However, research and innovation performers are aware and constantly draw upon societal needs, challenges, and public benefits to justify public sponsorship. At the same time, universities, public research organizations, and the private sector actively seek property rights over potential applications through patenting, which is consistent with the worldwide growth in patent filings over recent decades. Governments use patents systems to correct market failure and as a way to incentivize investment in research and development. This is based on the assumption that unprotected, free knowledge will bring low or null financial returns to inventors, which will affect investments in research and innovation, market productivity, and economic and social outcomes [66].

For this reason, a patent, in addition to other mechanisms such as prizes and research contracts, aims to incentivize research and development to attract investments and promote technological markets. The private value of a patent can be defined as the economic gains from exclusivity rights granted to an invention, although a patent can also have public value, as it encourages information sharing, furthers R+D investments, and inspires useful applications of emerging knowledge. In this way, the patent system is a socially shaped institution in which private and public concerns intersect. In this regard, the private value of conotoxin patents is reserved for Prialt-related companies, which have earned millions of dollars (~27 MD/year) [39]. However, the public value of conotoxins has risen to an unexpected level because of the many medical research tools that have emerged from them, as [9,29] mentioned, many conotoxins and conopeptides can be used as unique sources of diagnostic and research tools to study signaling pathways given their target specificity and in the particular case of ω-conotoxins, they have been used in more than 3000 published studies.

After analyzing the last six years of conotoxin patents, we noticed a solid tendency towards pharmaceutical applications, followed by uses in the biotechnological analysis of biological materials. This indicates that conotoxins are more valuable as pharma and biotech tools, as other uses have decreased overall, such as the analysis of biological materials (from 36 to 2 patents), basic materials chemistry (from 12 to 3), and medical (8 to none), which has now disappeared along with other minor uses.

The work in [29] anticipates that the combination of high-content target screening and the interest of the pharmaceutical industry in peptide-based drug development, will lead to the development and design of multiple conotoxin-based biomedical tools and pharmacological agents in the future. Furthermore, these authors noted that recent advances in throughput and the sensitivity of next-generation DNA and peptide sequencing have resulted in a massive increase in the rate of conotoxin discovery. Other authors also agree that the future of conotoxins relies on next-generation DNA and peptide sequencing techniques and the chemical optimization of the peptides [3,18,67]. However, this appears to be complicated, as a DNA sequence cannot be used to predict post-translational modifications, which are essential for conotoxin activity.

It is important to know which patents are currently enforced. Many assignees have submitted plenty of patent applications and could be considered to be the most relevant leaders; however, when analyzing the validity of their patent portfolio, we observed that the majority of patents had expired or were abandoned at the time. Currently, with regard to patent volume, the Chinese company BGI stands out as a leader with 13 granted patents between 2014 and 2015. It is possible that BGI will remained a force to be reckoned with until 2035 in places such as Switzerland, China, Germany, France, and the United States. Hainan University follows closely behind with eight granted valid patents, two pending patents, and three expired patents. Our analysis highlights that China has had the highest number of patent applications in recent years and also hosts the highest number of national inventors, whose patents could be valid until 2035.

We noticed a clear pathway in conotoxin patenting led by academia from the time of the first patent, and this tendency has been sustained over the years. This may be an objective indicator that only a few companies are interested in investing in conotoxin R+D. Durek and Craik [18], mention that conotoxins may have been launched too early into the market. Given the previous skepticism over peptides as drugs by big pharma, it is probably not surprising that more conotoxin drugs have not reached the market. Whereas small biotech companies have held a ‘belief’ in peptides, they have perhaps lacked the expertise and resources to commit to the drug development process and have taken relatively raw leads into clinical trials too early.

Conotoxins are not simple molecules, and their discovery depends on the technological abilities of inventors and academics rather than on the R+D section that a company may set up for this purpose. For example, Cognetix, a company dedicated to these developments, was eventually dissolved along with many other start-ups and spin-offs that have tried to make their way into the pharmaceutical market (including Xenome). This is clearly a difficult task to achieve, and it would probably be easier for academics to detect and begin working with these peptides (as Olivera and colleagues did) and to protect and patent their results. Then, these academics could explore the possibilities of transferring their patent portfolios to an interested company with the necessary infrastructure to develop a final product from a peptide and the experience to scale-up the process to industrial production, sales, and distribution. In addition, molecules could be developed within the academic sector but with the support of a pharmaceutical company from the outset.

Conotoxin research has not been easy endeavor. After more than 30 years of intensive global research and different companies that have invested in this research, only one successful pharmaceutical product has reached the market, despite years of effort and substantial resources that have been invested [18] and compared with products obtained from other marine organisms which are already in the market [13]. The main reason for this could be the lack of communication or aligned goals between the academic community and the private industry. This lack of success for the conotoxin patents, can take a positive turn in the near future, because a military organization, and a few companies are betting on conotoxin-based developments without the academic pressure to publish papers.

## 4. Materials and Methods

The current study globally analyzed conotoxin patent data and generated an overview of the patenting trends; primary applications; technological trends; current technological leaders based on active and granted patents; the companies and academic institutions involved in R+D; the number of live, lapsed, revoked, and abandoned patents; the countries/territories covered by patents filing; and the most relevant collaborations and licensing agreements of conotoxins and conopeptides. All data were obtained by using the web-based patent and design search software Orbit Intelligence (Questel), a useful patent data mining tool with access to a unique database comprising 125 million patent publications from 110 patent-issuing authorities (i.e., countries/territories and organizations). The database spans 91 million patents and 63 million patent families and returns duplicate-free results. The data coverage of Orbit includes full-text patent data from 59 countries/territories and organizations and 60 patent-issuing authorities with daily updates of the legal status and licensing information of 450,000 patents. These features allow for complete and reliable patent technology analyses, as has been done in other studies [19,68,69,70,71,72,73].

Orbit Intelligence includes high-quality data mining tools and databases with deep processing features that can be arranged according to functional requirements. This makes it possible to analyze individual databases with unique features and to then create clusters and merge these into individual databases with the previously selected patent documents. The resulting databases thus contain reliable, complete, and selected information.

The first searching strategy we designed was focused on the Title and Claims sections of Orbit Intelligence. This search matched the term “conotox+” in combination with all the IPC codes related to “Biotechnology” or “Pharmaceuticals,” as it was previously detected that these kinds of technologies represent the main uses for conotoxins. Then, a second search was performed with the term “conus?” instead of “conotox+” to ensure that all molecules and compounds derived from *Conus* snails were represented; the rest of the terms were maintained the same in the second search.

The results from the first searching strategy were 340 patents documents, but only 200 were aligned to the focus of this study. Additionally, from the second search, we obtained 147 hits, but only 24 were useful and not represented in the first search. Thus, 224 documents were considered relevant and used for the patent information analysis.

After checking twice for their relevance, 224 patent documents, which included granted patents and patent applications, were selected for full-text analysis. Document selection was conducted in a detailed and careful manner to avoid duplication. As such, each document represents a different simple patent family. The resulting document collection was thus composed of documents that had been carefully and purposefully examined and that prioritized the representativeness of different inventions. 

## Figures and Tables

**Figure 1 marinedrugs-20-00531-f001:**
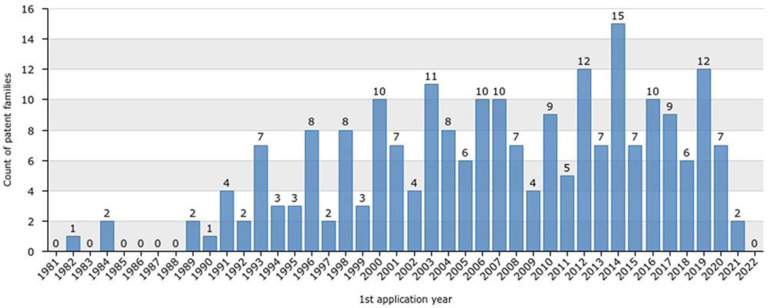
Conotoxin patent applications overview for the last 41 years.

**Figure 2 marinedrugs-20-00531-f002:**
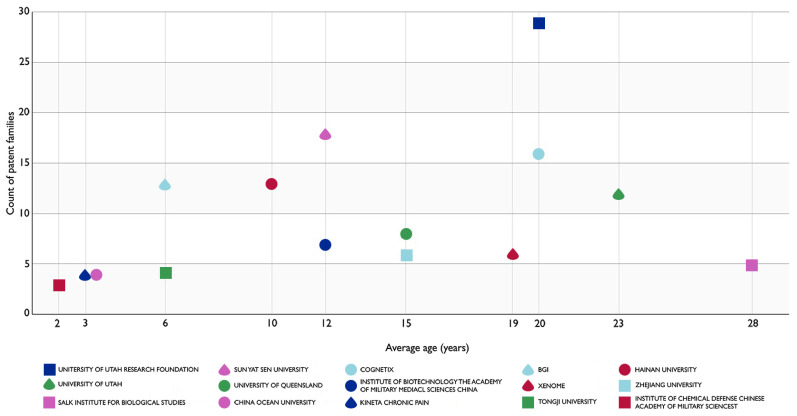
Assignee age in years by patent portfolio size. The 15 most relevant applicants are shown, from the pioneer (Salk Institute for Biological Studies) to the youngest assignee (Institute of Chemical Defense Chinese Academy of Military Sciences).

**Figure 3 marinedrugs-20-00531-f003:**
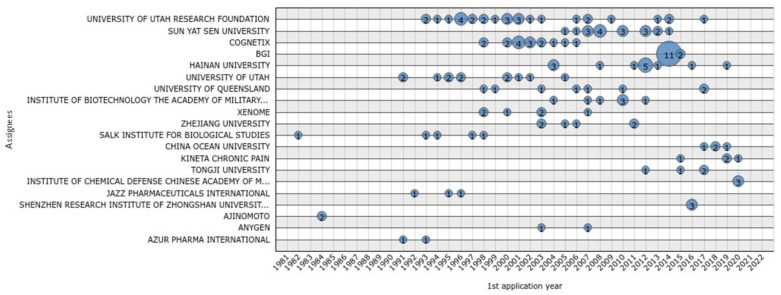
Top 20 assignees patent family number per year during the last 41 years.

**Figure 4 marinedrugs-20-00531-f004:**
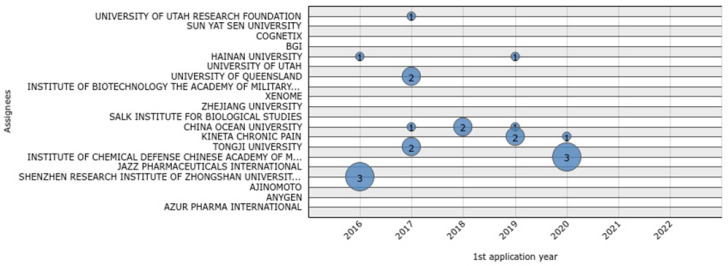
Patent applications from top 20 conotoxin assignees during the last 6 years.

**Figure 5 marinedrugs-20-00531-f005:**
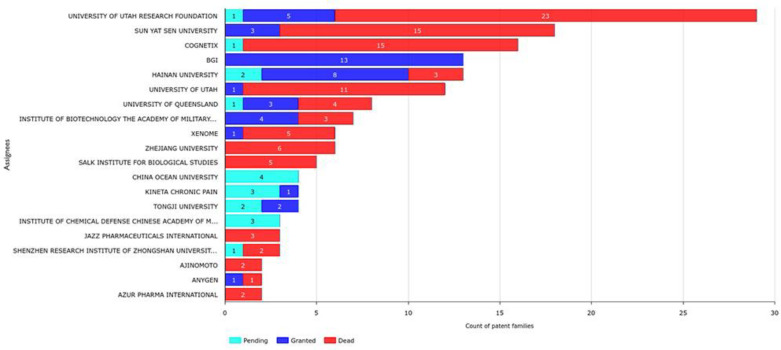
Patent legal status from top 20 conotoxin assignees.

**Figure 6 marinedrugs-20-00531-f006:**
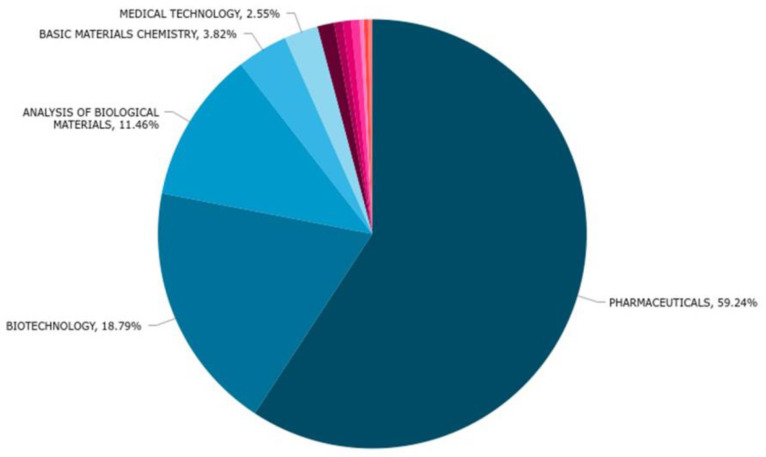
Technology domain of conotoxin patents from 1981 to June 2022.

**Figure 7 marinedrugs-20-00531-f007:**
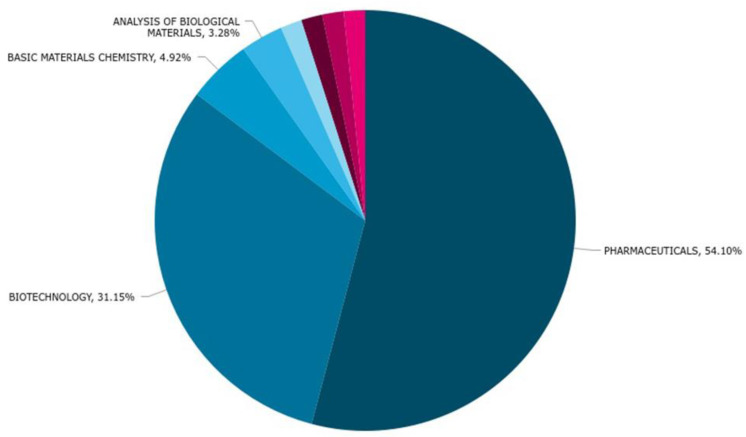
Technology domain of conotoxin patents during the last 6 years.

**Figure 8 marinedrugs-20-00531-f008:**
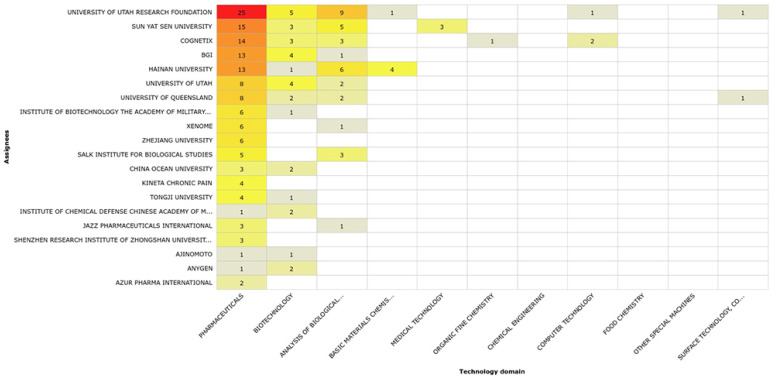
Technology domain of conotoxin patents by assignee. Red indicates the assignee with the largest number of patents, orange, yellow and beige indicate assignees with less patents.

**Figure 9 marinedrugs-20-00531-f009:**
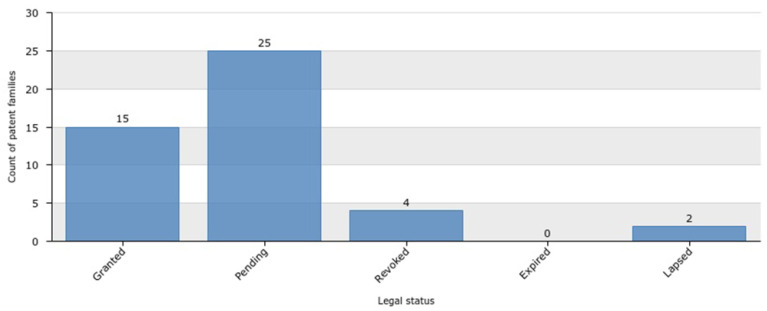
Legal status of conotoxin patent applications from 2016 to 2022.

**Figure 10 marinedrugs-20-00531-f010:**
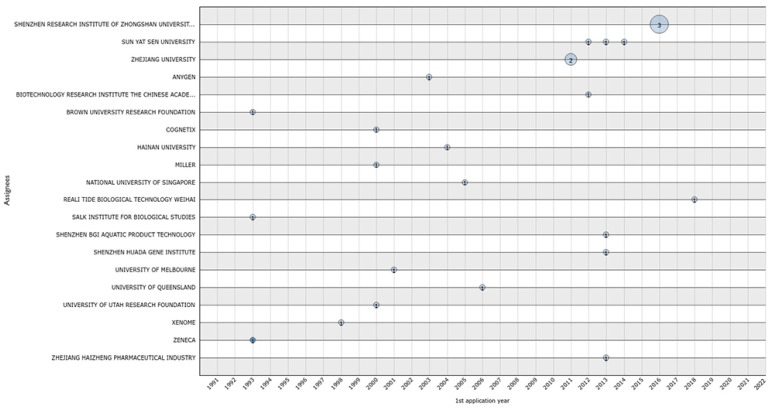
Revoked conotoxin patents from 1981 to 2022.

**Figure 11 marinedrugs-20-00531-f011:**
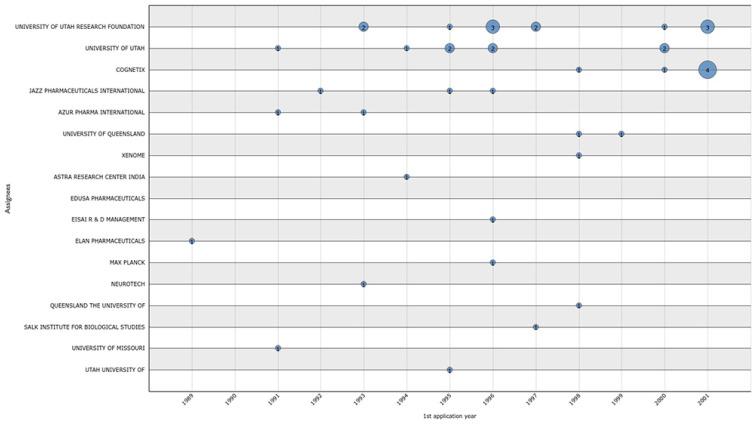
Expired conotoxin patents by year of application, from 1989 to 2001.

**Figure 12 marinedrugs-20-00531-f012:**
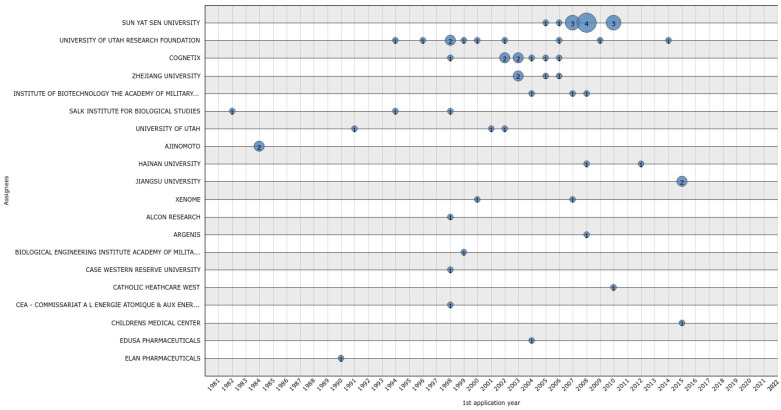
Lapsed conotoxin patents by year of application, from 1981 to 2022.

**Figure 13 marinedrugs-20-00531-f013:**
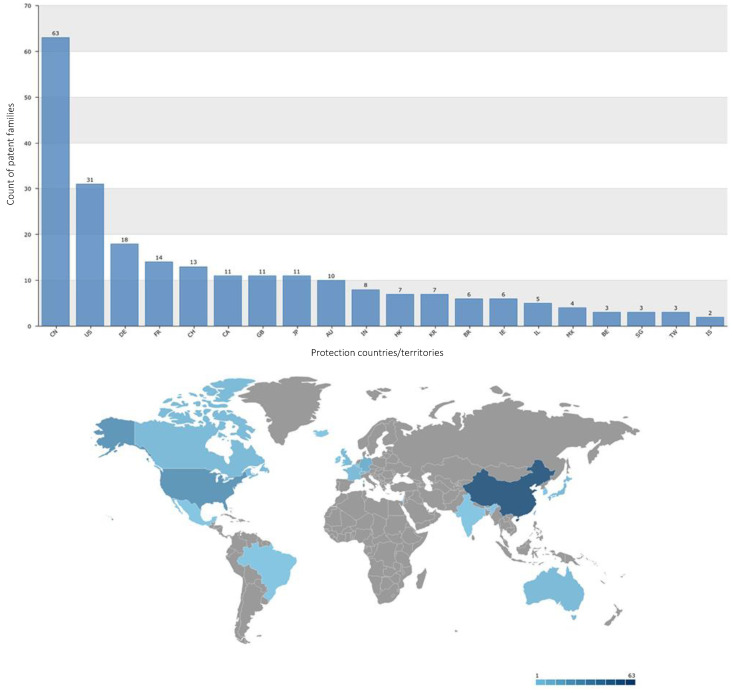
Top 20 countries/territories where the assignees have submitted conotoxin patent applications that are still valid.

**Figure 14 marinedrugs-20-00531-f014:**
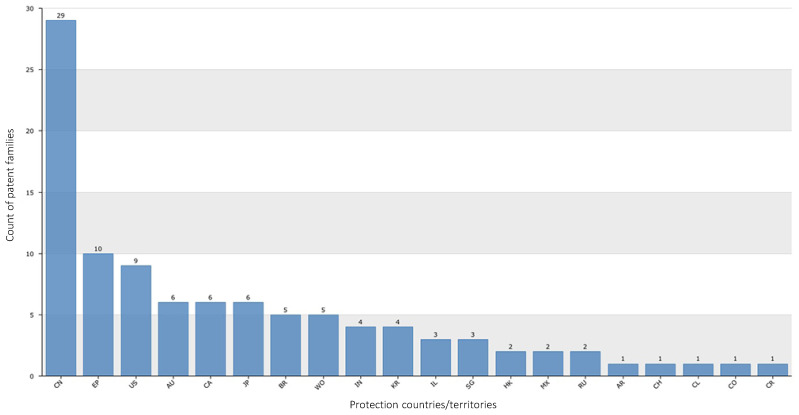
Top 20 countries/territories where the assignees have submitted conotoxin patent applications between 2016 and 2022.

**Figure 15 marinedrugs-20-00531-f015:**
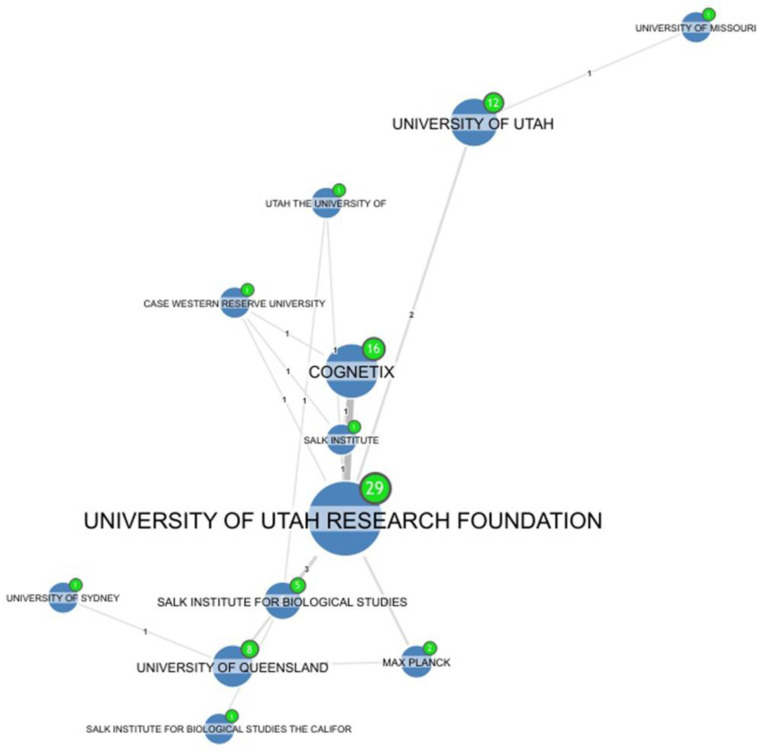
Collaborations related to the University of Utah Research Foundation and other patents assignees (Cognetix, Salk Institute, University of Queensland and others).

**Figure 16 marinedrugs-20-00531-f016:**
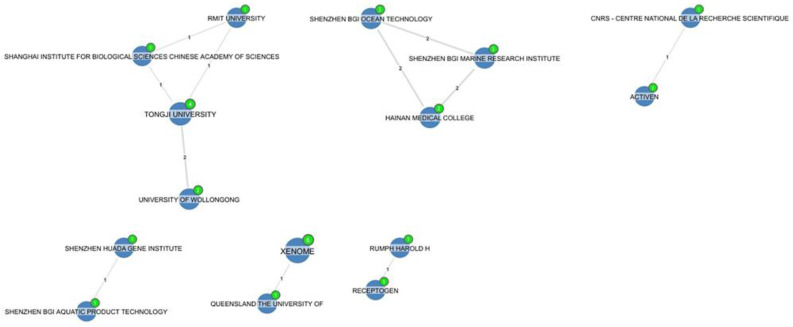
Collaborations between other conotoxin patents assignees.

**Figure 17 marinedrugs-20-00531-f017:**
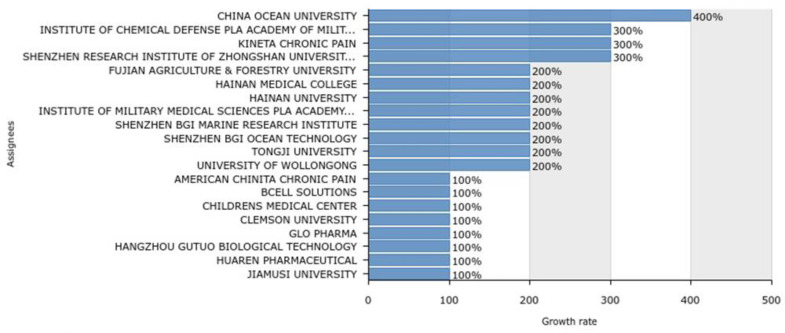
Assignees with the highest growth rates over the last six years (2016–2022), based on the number of filings per year.

**Figure 18 marinedrugs-20-00531-f018:**
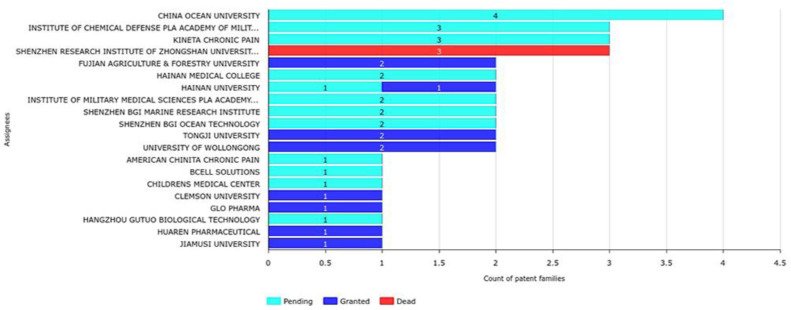
Top 20 assignees legal status from 2016 to 2022.

**Figure 19 marinedrugs-20-00531-f019:**
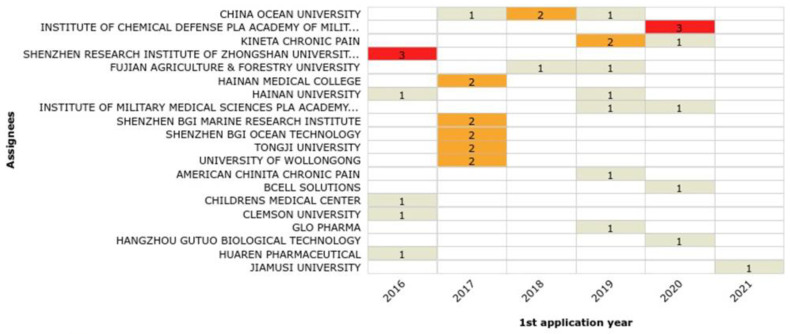
Top 20 assignees number of applications over the last six years (2016–2022). Red indicates the assignees with the largest number of patents, orange and beige indicate assignees with less patents.

## Data Availability

Not applicable.

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
