# Peer review of "Conotoxin Patenting Trends in Academia and Industry"

_marinedrugs, 2022, doi:10.3390/md20080531_

Round 1

Reviewer 1 Report

Conotoxins or conopeptides are small peptides from the venom of the Cone snails, and have great potential to be developed into drugs or pharmacological tools. In this manuscript, the author systematically reviewed 224 patent documents related to conotoxins and conopeptides to determine the course that innovation and development has taken over the years, their primary applications, the technological trends over the last five years, and the leaders in the field. This review is interesting and important in consideration of the great application potential of conopeptides. The only existing conotoxin patent review was performed in 2015 by Australian researchers from the IMB of Queensland. An updated review is essential to the field of conotoxin drug development. Thus, I recommend to publish the manuscript after minor revision.

 1.     In the abstract part a comment on the purpose or the implication of this review is required.

2.     Some inappropriate expression problems, such as “In addition, we explored which countries protect their inventions and patents 16 and the most relevant collaborations among actors.”, what “actors” means ?/” led to different patents being filed to try and improve its administration route and it effect when combined with other analgesics[16].” “it” should be changed into its ?(line 66)/ “analyzed to obtain quality information, it is necessary to use innovative patent mining (line 78)” “quality information” should be corrected/” Figure (2) is a good indicator of the level of inventiveness of all active players,”(line 129)/ Tongji University constitute a new, strong applicant group with 1- and 2-year-old portfo-137 lios that could active for up to 18 additional years (Line 138). “could active” is not correct/” One of the main objectives of this study was to detect active assignees with granted patents and to study the evolution of conotoxins during the last five years”, evolution of the conotoxins seems to be confusing here/ “The patents reflected only one use each of conotoxins in organic fine chemistry and coat-196 ing and surface technology, and their use in medical technology has disappeared com-197 pletely ” The sentence seems to be confusing./” In January 2000 in Australia, Dr. Craik, Dr. Richard Lewis, and Professor Alewood founded Xenome Ltd., a spin-off company from the University of Queensland (264)” “Dr. Lewis” ?

3.     “China Ocean University” should be changed into “Ocean University of China’   

4.     The different colors in Figure 8 and Figure 19 should be explained. What does the red and yellow color represents, respectively ?

Author Response

Conotoxins or conopeptides are small peptides from the venom of the Cone snails, and have great potential to be developed into drugs or pharmacological tools. In this manuscript, the author systematically reviewed 224 patent documents related to conotoxins and conopeptides to determine the course that innovation and development has taken over the years, their primary applications, the technological trends over the last five years, and the leaders in the field. This review is interesting and important in consideration of the great application potential of conopeptides. The only existing conotoxin patent review was performed in 2015 by Australian researchers from the IMB of Queensland. An updated review is essential to the field of conotoxin drug development. Thus, I recommend to publish the manuscript after minor revision.

  1. In the abstract part a comment on the purpose or the implication of this review is required.

Answer: We correct the wrong used terms in the abstract section, and we also changed "We noticed" by "We concluded", so the abstract can transmit what we found during the patent review we performed, considering that conotoxin patents have more academic than industrial value based on the number of licensing and contributions to medical research. We also included the aim of this review at the end of the abstract. With those changes, we consider that now the abstract is clearer about the purpose and implication of this review (determine the course that innovation and development has taken over the years, primary applications, technological trends over the last five years and the leaders in the field of conotoxins), and we also denoted that there is only one previous patent review, from 2015.

  1. Some inappropriate expression problems, such as “In addition, we explored which countries protect their inventions and patents 16 and the most relevant collaborations among actors.”, what “actors” means ?

Answer: We changed the term "actors" for the right term, which is "assignees".

” led to different patents being filed to try and improve its administration route and it effect when combined with other analgesics[16].” “it” should be changed into its ?(line 66)

Answer: We corrected the misspelling, resulting in "has led to different patents being filed to try to improve its administration route and its effect when combined with other analgesics [16]."

“analyzed to obtain quality information, it is necessary to use innovative patent mining (line 78)” “quality information” should be corrected/”  

Answer: "quality information" was corrected to "reliable information".

Figure (2) is a good indicator of the level of inventiveness of all active players,”(line 129)/ Tongji University constitute a new, strong applicant group with 1- and 2-year-old portfo-137 lios that could active for up to 18 additional years (Line 138). “could active” is not correct/”

Answer: Corrected to "all the active players" in line 129, since we are referring to the assignees with more patents which could be active during the next years if the assignees decide to pay all the future coming patens fees until patents expire. We also corrected, "could active" with "could be active" in line 138.

One of the main objectives of this study was to detect active assignees with granted patents and to study the evolution of conotoxins during the last five years”, evolution of the conotoxins seems to be confusing here/

Answer: Totally agree, we add the term "evolution of patent conotoxins" in this and other lines to make clear that we want to refer to the patents along the years and not to the conotoxins (peptides) evolution as well.

“The patents reflected only one use each of conotoxins in organic fine chemistry and coat-196 ing and surface technology, and their use in medical technology has disappeared com-197 pletely ” The sentence seems to be confusing./

Answer: The English editor who reviewed this manuscript first considered a better option to change this sentence. Now we chose to let it as it was before and it reads as follows: "The patents claims reflected only one use of conotoxins in organic fine chemistry and one use in coating and surface technology, and their use in medical technology has now disappeared completely when compared to the patents filed in the past (Figure 7)."

” In January 2000 in Australia, Dr. Craik, Dr. Richard Lewis, and Professor Alewood founded Xenome Ltd., a spin-off company from the University of Queensland (264)” “Dr. Lewis” ?

Answer: Totally agree. This was noted also by another reviewer, so it was corrected to "In January 2000 in Australia, Dr. David Craik, Dr. Richard Lewis, and Professor Paul Alewood founded Xenome".

  1. “China Ocean University” should be changed into “Ocean University of China’   

Answer: China Ocean University it is the exact assignee/applicant name cited on the patents along the manuscript, but we changed it for "Ocean University of China" to become more clear about which University we want to refer to.

  1. The different colors in Figure 8 and Figure 19 should be explained. What does the red and yellow color represents, respectively ?

Answer: We already have included a brief explanation of the color scale in figure 8… "Red indicates the assignee with the largest number of patents, orange, yellow and beige indicate assignees with less patents.", and in Figure 19 "Red indicates the assignees with the largest number of patents, orange and beige indicate assignees with less patents".

Reviewer 2 Report

The manuscript by Sanchez-Campos et al. provides an updated overview of the conotoxin patenting trends in academia and industry. This is a well written paper, which should be of interest to people working in the field as well as those interested in patent outcomes from natural products in general. I have only minor comments:

- The figures, as I understand it, were automatically generated by the program Questel. Unfortunately, many of these figures are unreadable because of the small font used. I would suggest the authors to import these original figures into an image software where they can modify the font. The choice of colors is also poor in some cases, as the subtle variations of blue or pink are difficult to tell apart (i.e. Fig 2).

- Many researchers in the field might not be familiar with some of the terms used. I would suggest to add the definition of some of these specific terms like "lapsed patents", "abandoned patents", "expired patents", as they are not common jargon in science.

- I feel that the discussion is lacking some comparative perspectives. For instance, when the authors state that "the field of conotoxins does not seem to be very successful at all", without a comparison to "more successful fields", this statement is purely subjective and does not really help. How does conotoxin patents fare compare to similar products (peptides from other natural sources)? Are they really less successful? If so, the authors should demonstrate it. A comparative figure would be useful.

- Page 17, lines 504 to 507 belongs to the results section.

- page 1, l40: "prove" is not the appropriate term for what the authors want to say. Maybe "use" or "test" would fit better here.

- page 2, l82: some informations are missing. The authors "only focused in analysing 224 patent documents", and dismissed 137 other related patents. It is not clear what criteria were used to dismiss these ("not including any specific innovation related to novel conotoxin use" is vague....).

- page 3, l111, please remove the coma after "two".

- page 4, l124: "BGI" appears for the first time in the text, please write in full or add an abbreviation list at the beginning of the paper.

- page 5, l161: "the evolution of conotoxin" is confusing, as this is a field of research on its own. The authors likely meant "the evolution of conotoxin related patents during the last five years".

- page 10, l293 "mentioned" not mention.

Author Response

The manuscript by Sanchez-Campos et al. provides an updated overview of the conotoxin patenting trends in academia and industry. This is a well written paper, which should be of interest to people working in the field as well as those interested in patent outcomes from natural products in general. I have only minor comments:

- The figures, as I understand it, were automatically generated by the program Questel. Unfortunately, many of these figures are unreadable because of the small font used. I would suggest the authors to import these original figures into an image software where they can modify the font. The choice of colors is also poor in some cases, as the subtle variations of blue or pink are difficult to tell apart (i.e. Fig 2).

Answer: All the figures were imported to Photoshop where they were modified to assure that they are understandable and readable, also all the colors in figure 2 were modified and we add different bullet types for this figure, to make it more clear.

- Many researchers in the field might not be familiar with some of the terms used. I would suggest to add the definition of some of these specific terms like "lapsed patents", "abandoned patents", "expired patents", as they are not common jargon in science.

Answer: We agree so we add a sentence to explain each case and be clearer about the legal status in each case.  "2.4. Legal status of patent applications. From 1981 to 2022, we found 90 active (alive) conotoxin patents and 134 that were already dead (inactive). Of the 90 live patents, 60 had been granted and 30 were still pending. Of the 134 dead or inactive patents, 23 had been revoked (invalidated patent and cancelled rights), 34 had expired (patents are valid for up to 20 years since their first filing application date), and 77 had lapsed (anticipated expiration due to non payment)."

- I feel that the discussion is lacking some comparative perspectives. For instance, when the authors state that "the field of conotoxins does not seem to be very successful at all", without a comparison to "more successful fields", this statement is purely subjective and does not really help. How does conotoxin patents fare compare to similar products (peptides from other natural sources)? Are they really less successful? If so, the authors should demonstrate it. A comparative figure would be useful.

Answer: We agree, so we add a sentence in the discussion regarding this issue and cited an article, so it reads like this: " Conotoxin research has not been easy endeavor. After more than 30 years of intensive global research and different companies that have invested in this research, only one successful pharmaceutical product has reached the market, despite years of effort and substantial resources that have been invested [18] and compared with products obtained from other marine organisms which are already in the market [13]."

- Page 17, lines 504 to 507 belongs to the results section.

Answer:  We appreciate your suggestion regarding this section. In this particular case, we consider that those lines belong and fit well in the materials and methods section, since it describes the searching strategy that was followed to perform the patent searching.

- page 1, l40: "prove" is not the appropriate term for what the authors want to say. Maybe "use" or "test" would fit better here.

Answer: We changed "prove" for "test".

- page 2, l82: some informations are missing. The authors "only focused in analysing 224 patent documents", and dismissed 137 other related patents. It is not clear what criteria were used to dismiss these ("not including any specific innovation related to novel conotoxin use" is vague....).

Answer: To avoid vague sentences or affirmations we modified this sentence:   "Although these 137 patents included conotoxins, they did not include any specific innovation related to novel conotoxin use, since they were used as research tools to test their inventions which were not related to new and relevant conotoxins uses per se." expecting that with this modofocation, it will become more clear that we focus our patent search in conotoxins uses not as helping tools, but as new and relevant inventions.

- page 3, l111, please remove the coma after "two".

Answer: We removed the coma.

- page 4, l124: "BGI" appears for the first time in the text, please write in full or add an abbreviation list at the beginning of the paper.

Answer: We add the full name of the company BGI the first time it appears in the text: "A 4-year gap can then be observed, which is followed by the 6-year-old portfolios of BGI/Beijing Genomics Institute".

- page 5, l161: "the evolution of conotoxin" is confusing, as this is a field of research on its own. The authors likely meant "the evolution of conotoxin related patents during the last five years".

Answer: Totally agree, we add the term "evolution of patent conotoxins" to make clear that we want to refer to the patents along the years and not to the conotoxins (peptides) evolution as well.

- page 10, l293 "mentioned" not mention.

Answer: We corrected this to "mentioned".

Reviewer 3 Report

The manuscript by Sanchez-Campos and co-workers presents an overview of the current state of patents and intelectual property protection regarding conotoxins and their potential medical applications. The central subject of this revision is not a scientific topic by itself, but it address most important issues concerning the use of conotoxins and the path from the laboratories to companies and ultimately society. The manuscript is well written and presented, and it is supported by a quite complete collection of bibliographic references. Given the fact that most of the introductory parts of articles on conotoxins state one way or the other their importance as  potential drug sources, this article presents a clear vision on what is the actual vision of the situation, as only one or two drugs derived from cone snail venom have make their way to the market. I consider therefore that the manuscript is of interest to readers working on conotoxins, and I recommend publication. I only have a couple of comments to make, for author's consideration:

1) The article refers repeatedly to genus Conus, and toxins derived from the venom of Conus, but nowadays it is well established the existence of several lineages associated with genera names (Californiconus, Profundiconus, Conasprella, Pygmaeconus, etc). It is true that most patents (if not all) come from Conus species, but perhaps the text might be slightly modified in order to accomodate the other lineages, which also have toxins .

2) Line 264: It states"Dr. Craik, Dr. Richard Lewis, and Professor Alewood". I believe that these names should be cited in a more formal and consistent manner, i.e. Dr. David Craik, Dr. Richard Lewis and Professor Paul Alewood". 

3) Finally, I see that the coverage of patent claims and their history is quite well documented. However I miss some information regarding the status of the several cone-derived drugs that started clinical trials. Actually only Ziconotide completed the different stages, whereas the other ones stopped at different stages of clinical trial for different reasons, and development was discontinued. Although this is not strictly related to patents, I believe that a small table providing information about these drug candidates and how they performed in clinical trials should be of utmost interest for readers. 

Author Response

The manuscript by Sanchez-Campos and co-workers presents an overview of the current state of patents and intelectual property protection regarding conotoxins and their potential medical applications. The central subject of this revision is not a scientific topic by itself, but it address most important issues concerning the use of conotoxins and the path from the laboratories to companies and ultimately society. The manuscript is well written and presented, and it is supported by a quite complete collection of bibliographic references. Given the fact that most of the introductory parts of articles on conotoxins state one way or the other their importance as  potential drug sources, this article presents a clear vision on what is the actual vision of the situation, as only one or two drugs derived from cone snail venom have make their way to the market. I consider therefore that the manuscript is of interest to readers working on conotoxins, and I recommend publication. I only have a couple of comments to make, for author's consideration:

1) The article refers repeatedly to genus Conus, and toxins derived from the venom of Conus, but nowadays it is well established the existence of several lineages associated with genera names (Californiconus, Profundiconus, Conasprella, Pygmaeconus, etc). It is true that most patents (if not all) come from Conus species, but perhaps the text might be slightly modified in order to accomodate the other lineages, which also have toxins.

Answer: Thank you for denoting this, we are aware of this lineages, so the patent search we performed was planned to cover all the lineages associated with Conus and we did not found any patent related yet to Californiconus, Profundiconus, Conasprella, Pygmaeconus, or others, so we keep it just like Conus and conotoxins/conopeptides.

2) Line 264: It states"Dr. Craik, Dr. Richard Lewis, and Professor Alewood". I believe that these names should be cited in a more formal and consistent manner, i.e. Dr. David Craik, Dr. Richard Lewis and Professor Paul Alewood". 

Answer: This was noted also by another reviewer, so it was corrected to " In January 2000 in Australia, Dr. David Craik, Dr. Richard Lewis, and Professor Paul Alewood founded Xenome".

3) Finally, I see that the coverage of patent claims and their history is quite well documented. However I miss some information regarding the status of the several cone-derived drugs that started clinical trials. Actually only Ziconotide completed the different stages, whereas the other ones stopped at different stages of clinical trial for different reasons, and development was discontinued. Although this is not strictly related to patents, I believe that a small table providing information about these drug candidates and how they performed in clinical trials should be of utmost interest for readers. 

Answer: We add a sentence in part 2.6 Companies interested in conotoxins, that will guide the reader to a pair of reviews, which contains this information in a more detail manner.    "An updated overview regarding conotoxins as drug leads and their current status in clinical trials or development stage is well documented in [29] and up to 2015 in [13]."